# Antibiotic Resistance Profiles in Eye Infections: A Local Concern with a Retrospective Focus on a Large Hospital in Northern Italy

**DOI:** 10.3390/microorganisms12050984

**Published:** 2024-05-14

**Authors:** Lorenzo Drago, Vincenzo Minasi, Andrea Lembo, Angela Uslenghi, Sofia Benedetti, Matteo Covi, Paolo Nucci, Loredana Deflorio

**Affiliations:** 1UOC Laboratory of Clinical Medicine with Specialized Areas, IRCCS MultiMedica, Via Fantoli 16/15, 20138 Milan, Italy; vincenzo.minasi@multimedica.it (V.M.); angela.uslenghi@multimedica.it (A.U.); sofia.benedetti@multimedica.it (S.B.); matteo.covi@multimedica.it (M.C.); loredana.deflorio@multimedica.it (L.D.); 2Clinical Microbiology and Microbiome Laboratory, Department of Biomedical Sciences for Health, University of Milan, Via Mangiagalli 31, 20133 Milan, Italy; 3Department of Biomedical, Surgical and Dental Sciences, University Eye Clinic, San Giuseppe Hospital-IRCCS MultiMedica, University of Milan, Via San Vittore 12, 20123 Milan, Italy; andrea.lembo@multimedica.it (A.L.); paolo.nucci@unimi.it (P.N.)

**Keywords:** antibiotic resistance, eye infections, bacteria, risk factors

## Abstract

The emergence of antibiotic resistance poses a significant threat to public health worldwide, affecting various medical fields, including ophthalmology. Eye infections, ranging from conjunctivitis to more severe conditions like keratitis, are commonly treated with antibiotics. However, the misuse and overuse of these drugs have led to the development of resistant strains of bacteria, allowing traditional treatments ineffective. This paper aims to examine the current situation of antibiotic resistance in eye infections globally, with a specific focus on a large group of hospitals located in Milan (Italy) with considerable experience in cataract and cornea surgery as well as in retinopathy. The results of the study show the prevalence of Gram-positives in the tested samples and a low resistance of fluoroquinolones and glycopeptides. The results also highlight the need to implement sample collection methods for ocular infections, as the quantity of positive samples is rather low compared to the total number of samples. In conclusion, the study, although with limited data, shows that resistance to aminoglycosides and cephalosporins is a situation to be monitored. These data also show the critical need to improve and guide the biological sample collection modalities in order to make the diagnosis more reliable.

## 1. Introduction

Antibiotic resistance in eye infections has become a growing concern globally. The indiscriminate use of antibiotics, both in clinical and over-the-counter settings, has accelerated the development of resistant strains of bacteria. According to the World Health Organization (WHO), antibiotic resistance is now a significant threat to global health security, with estimates suggesting that by 2050, it could cause 10 million deaths annually if left unchecked [1]. In the field of ophthalmology, bacterial conjunctivitis, keratitis, and endophthalmitis are among the most common infections encountered, and their management is becoming increasingly challenging due to antibiotic resistance [2]. 

In recent years, several studies have highlighted the prevalence of resistant bacterial strains in ocular specimens. A study conducted by Collier et al. [3] in the USA found that among patients diagnosed with bacterial keratitis, approximately 40% showed resistance to commonly used antibiotics such as fluoroquinolones. Italy, like many other countries, is not immune to the challenges posed by antibiotic resistance in eye infections. A surveillance study by an Italian group reported a concerning increase in the prevalence of aminoglycosides and methicillin-resistant staphylococci (MRSA) in cases of eye infections in a 30-year retrospective analysis in the urban area of Turin [4,5,6,7].

Several factors contribute to the development and spread of antibiotic resistance in eye infections. One of the primary drivers is the inappropriate use of antibiotics, including self-medication and inadequate dosing or duration of treatment. Inadequate infection control practices in healthcare settings can also facilitate the transmission of resistant strains among patients [8]. 

Inappropriate use is not only attributable to the patient, but also sometimes to the medical specialist, who, out of some sort of legal medical fear, tends to prescribe antibiotics with a broad spectrum of action. Another interpretation of this can be given by the overcrowding of clinics, which leads the doctor to prescribe a more “comfortable” antibiotic molecule to avoid new access by the patient for what he considers a minor infection. Furthermore, the widespread use of antibiotics in agriculture and animal husbandry contributes to the environmental reservoir of resistant bacteria, which can subsequently infect humans through various routes, including contaminated water sources [9].

Patient risk factors, such as diabetes or diabetic retinopathy, put patients at an increased risk of developing conjunctival and corneal bacterial infections, including acute infectious conjunctivitis and keratitis [10]. 

Addressing antibiotic resistance in eye infections requires a multifaceted approach involving healthcare professionals, policymakers, and the general public. First and foremost, there is a need for improved surveillance of antibiotic resistance patterns in ocular pathogens to guide empirical treatment decisions. Healthcare providers must adhere to rational prescribing practices, ensuring that antibiotics are used judiciously and only when necessary. Patient education campaigns can raise awareness about the risks of antibiotic overuse and the importance of completing prescribed courses of treatment to prevent the development of resistance.

In addition, infection prevention and control measures, such as hand hygiene and proper disinfection of medical instruments, are crucial for reducing the spread of resistant bacteria in healthcare settings. The increasing use of contact lenses in the population, given the exponential increase in myopia worldwide, has also increased the incidence of corneal infections and the spread of saprophytic germs to the hydrogel silicone used in the manufacture of contact lenses. Furthermore, there is a need for continued research and development of novel antimicrobial agents, including alternative therapies such as bacteriophages and antimicrobial peptides, which may offer new treatment options for antibiotic-resistant infections [11].

Antibiotic resistance in eye infections represents a significant public health challenge, with implications for both individual patient outcomes and healthcare systems as a whole. The situation is particularly concerning in Italy, where high rates of resistance have been reported in ocular pathogens [4]. Addressing this issue requires a concerted effort that implements effective infection control measures by evaluating the epidemiological situation of the most-tested antibiotics in vitro and by investing in the research and development of alternative treatment strategies. By taking these steps, we can mitigate the impact of antibiotic resistance and ensure the effective management of ocular infections in the future. 

At the national level, surveillance programs monitor antibiotic resistance patterns across different regions and healthcare settings, allowing health authorities to identify emerging resistance trends and prioritize resources for interventions. For example, the Centers for Disease Control and Prevention (CDC) in the United States maintains the National Healthcare Safety Network (NHSN) to track antibiotic resistance in healthcare-associated infections, providing data to inform infection prevention strategies and antimicrobial stewardship programs [12]. Understanding the antibiotic profile of pathogens at various geographic levels, such as national, regional, or local, is crucial for the effective management of infectious diseases and combating antibiotic resistance. This knowledge provides valuable insights into the prevalence of resistant strains, guiding healthcare providers in selecting appropriate antimicrobial therapies and implementing targeted interventions to control the spread of resistance. For all these reasons, the aim of this work was to see a 6-year epidemiological snapshot, with a particular focus on antibiotic resistance in our hospital group, which includes many patients with ocular issues attending our eye divisions.

## 2. Materials and Method

A retrospective cross-section analysis, approved by the Internal Ophthalmologists Board, was conducted from 2018 to 2023 on all eye samples received in the laboratory throughout the analyzed period. The laboratory of microbiology examined a total of 267 specimens, coming from patients with secretion or pus production from eyes.

The eye samples were collected by swabs with Amies transport medium to carry out a culture examination for searching common microorganisms, including fungi.

The swabs were performed on patients in the various hospital facilities at the IRCCS MultiMedica Group, mainly from pediatric and adult ophthalmology departments, as well as by outpatient ambulatory settings.

The samples were analyzed by a culture method. In particular, Blood Agar, Mannitol Salt Agar, and McConkey agar were used for Streptococcus, Staphylococcus, and Enterobacteria/Pseudomonas isolation, respectively, while plates of Sabouraud Dextrose Agar (SDA) were used for fungi isolation. Plates were incubated for 24–48 h and for 72 h for bacteria and fungi, respectively. Isolates were subsequently processed for biochemical identification, and the susceptibility tests were carried out by means of Vitek2 Compact (BioMerieux, Marcy L’Etoile, France). For each isolate, the Minimal Inhibitory Concentrations (MICs) were evaluated according to the EUCAST for the antibiotic panels already pre-established by the VITEK2 system. It was not possible to test chloramphenicol as this antibiotic is not included into the antibiotic panels commonly used in laboratories.

## 3. Results

The results on the bacterial isolates show that the positive swabs were 54 on 267 collected specimens (20% of the total samples).

In particular, 39 Gram-positives (mainly *Staphylococcus aureus*, Coagulase-negative Staphylococci, *Streptococcus pneumoniae*, and *Streptococcus* spp.) and 15 Gram-negatives (mainly *Pseudomonas aeruginosa* and *Enterobacteriaceae*) were isolated (Table 1). As can be seen from Table 1, there was no report of extended-spectrum beta-lactamase (ESBL) positivity among the *Enterobacteriaceae*, and the incidence of methicillin resistance in *Staphylococcus aureus* (MRSA) was relatively low (4.5%). No methicillin-resistant strain was found in the six Coagulase-negative Staphylococci isolates. In addition, no yeasts were isolated during the period analyzed.

Regarding the antibiotic susceptibility profiles, the cumulative resistance to the tested glycopeptides and fluoroquinolones was low, 2.7% and 5.6%, respectively, while worthy of attention was the resistance found to aminoglycosides (17.9%), but above all to cephalosporines (25%) (Table 2). Regarding Gram-negatives, especially Enterobacteria, the only noteworthy resistances observed were those to amoxicillin (18%).

## 4. Discussion

Regionally, variations in antibiotic resistance profiles may exist due to differences in antimicrobial prescribing practices, healthcare infrastructure, and population demographics. Moreover, the frequency of certain infections in a specific population is a condition that leads doctors to prescribe a particular class of antibiotics. By assessing regional data, healthcare providers can tailor treatment guidelines and antimicrobial stewardship efforts to address specific local challenges. For instance, a study conducted in Europe found significant variability in antibiotic resistance rates among different regions, emphasizing the importance of region-specific surveillance to guide empirical therapy decisions [13].

At the local level, understanding antibiotic resistance patterns within individual healthcare facilities is essential for implementing targeted interventions to prevent healthcare-associated infections and optimize patient outcomes. Local surveillance data enable healthcare providers to identify outbreaks, implement infection control measures, and adjust empiric antibiotic therapy protocols based on local resistance profiles. For example, a study conducted in a tertiary-care hospital demonstrated the effectiveness of implementing a hospital-specific antimicrobial stewardship program in reducing antibiotic resistance rates and improving patient outcomes [14].

Our study shows a prevalence of Gram-positive isolations and good activity of fluoroquinolones and glycopeptides, which is quite similar to a study conducted in a tertiary referral hospital in the south of Italy [7]. In addition, we observed a growing resistance to aminoglycosides and third-generation cephalosporines. Among Gram-negatives, noteworthy is the resistance to amoxicillin of Enterobacteria. 

For many years, ophthalmologists used to prescribe large amounts of aminoglycosides, and the resistance to these molecules detected in recent years has pushed research toward a new generation of antibiotics or has brought back classes of drugs that have not been used for several years [15]. 

In light of these preliminary results, there exist certain challenges that impede optimal antibiotic selection, particularly in the case of ocular infections. Chloramphenicol is an overused agent in the antimicrobial armamentarium despite its absence from routine antibiotic panels. The dearth of data on chloramphenicol susceptibility limits the ability of clinicians to assess its efficacy in treating infections, thus undermining antibiotic stewardship efforts. Integrating chloramphenicol into routine antibiograms would provide clinicians with valuable insights into its susceptibility patterns and enable evidence-based antibiotic selection, thereby enhancing antimicrobial stewardship [16,17]. Despite its broad use against various pathogens, chloramphenicol still remains conspicuously absent from routine antibiograms. 

Although the majority of patients included in our study showed clinical signs of bacterial infections and justified the use of microbiological examination to confirm infection and find the clinical isolate, the high number of negative samples is also noteworthy.

Ocular infections present unique challenges in sample collection, often resulting in false-negative results, particularly with conventional swab-based techniques. Keratitis, characterized by inflammation of the cornea, poses a significant diagnostic challenge due to inadequate sample collection. The limited microbial load and diverse etiology of keratitis demand more sensitive diagnostic methods beyond conventional swabs [18]. Moreover, the reliance on clinical judgment alone for antibiotic selection in the absence of microbiological data compromises antibiotic stewardship efforts, leading to empiric therapy and potential overuse of broad-spectrum antibiotics. This aspect is limiting the clinical experience, which slows the ophthalmologist toward a wider reasoning and which floods his reins into semeiotics, without a deep analysis of the microbiological and clinical characteristics of a local infection (such as unilaterality, the presence of eye redness, the presence of conjunctival secretions, the quality of conjunctival secretions, and a well conducted microbiological diagnosis).

To overcome the limitations of conventional swab-based sample collection, alternative methods must be explored. Techniques such as corneal scrapings or aqueous humor aspiration offer a higher microbial yield and increased diagnostic accuracy in ocular infections, including keratitis [19]. 

Additionally, the integration of molecular diagnostic tests, such as polymerase chain reaction (PCR) and Next Generation Sequencing (NGS), holds promise in enhancing the rapid and accurate identification of pathogens in ocular samples. Molecular tests and metagenomics not only offer superior sensitivity and specificity but also enable targeted antibiotic therapy based on pathogen identification, thereby optimizing antibiotic stewardship [20].

Our study, albeit with limited data, gives an overview of the local situation, but above all highlights how much there is still to do in improving the quality of isolations, as well as the methods of collecting samples. The study certainly has some limitations, including the few positive samples (although it is not always easy to have large case histories in the various hospitals regarding ocular infections) and the fact that it is retrospective, where some clinical data are missing, in particular, a specific case history of where the pathologies come from.

Effective antibiotic stewardship is contingent upon accurate susceptibility data and a precise diagnosis of the infectious agents. A panel of antibiotics should be improved from ocular microorganisms (chloramphenicol should be included in order to warrant an ideal antibiogram to facilitate evidence-based antibiotic selection). The collaboration between specialists must be a firm point in clinical practice: in the case of doubts or infections that mimic other infections or that may have different etiology, the comparison among ophthalmologists, radiologists, and microbiologists may be decisive, increase knowledge, and lead to the collection of a more suitable sample. Moreover, addressing the challenges associated with sample collection in ocular infections, particularly keratitis, is crucial for enhancing diagnostic accuracy and guiding appropriate antibiotic therapy. Alternative sample collection methods and molecular tests offer promising avenues for improving diagnostic precision and optimizing antibiotic stewardship efforts in ocular infections.

In conclusion, knowledge of antibiotic profiles at national, regional, and local levels is critical for guiding antimicrobial therapy decisions. Our study, although with limited data, shows that Gram-positives are mainly responsible for ocular infections, followed by Gram-negatives. Resistance to aminoglycosides and cephalosporins is a situation to keep under control. The study also critically demonstrates the need to improve and guide the biological sample collection modalities in the different ocular districts.

## Figures and Tables

**Table 1 microorganisms-12-00984-t001:** Gram-positives and Gram-negatives microrganisms (number and percentage of MRSA and ESBL).

Gram-Positives	MRSA. N°/Total	% MRSA (*S. aureus*)	Gram-Negative	ESBLs (*Enterobacteriaceae*). N°/Total	% ESBLs
39	1/22	4.5	15	0/8	0

**Table 2 microorganisms-12-00984-t002:** Percentage of antibiotic resistance in the eye isolates.

% Resistance to Third Cephalosporines: All *S. pneumoniae*	25.0%
% Resistance to Fluoroquinolones: All Staphylococci and Streptococci	5.6%
% Resistance to Aminoglycosides/All Staphylococci	17.9%
% Resistance to Glycopeptides/Gram-Positives	2.7%
% Resistance to Amoxicillin/Enterobacteria	18%

## Data Availability

Data are contained within the article.

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
