# Peer review of "Antibiotic Resistance Profiles in Eye Infections: A Local Concern with a Retrospective Focus on a Large Hospital in Northern Italy"

_microorganisms, 2024, doi:10.3390/microorganisms12050984_

Round 1

Reviewer 1 Report

Comments and Suggestions for Authors

Thank you very much for allowing me to review this work. This is a cross-sectional study on ocular samples analyzed by microbiology from 2018 to 2023. The high prevalence of positive samples with cephalosporin-resistant bacteria is very striking.

I would like to provide several comments in order to contribute to improving the quality of the manuscript presentation.

1.- Abstract: Ok. Good estructured.

2.- Introduction. It should be reviewed. There are many sentences that are not referenced. It must be completed with references.

3.- Methods: The study design must be corrected. This is a cross-sectional study. The design must be clear. The power that the total samples analyzed contribute to the results should be reported.

4.- Results: Ok.

5.- Discussion: The first 3 paragraphs are more like writing the introduction than the discussion. This information should be eliminated or discussed focusing on the results. As in the introduction, many sentences appear without bibliographic support. It should be reviewed.

6.- Conclusions: The conclusion they provide is not supported by the results. It should be reformulated.

Author Response

Dear Reviewer, many thanks for your important inputs. You may find in the attached file all the answers (in bold and yellow).

With my best regards.

Prof. Lorenzo Drago

Reviewer 2 Report

Comments and Suggestions for Authors

The communication article  " Antibiotic Resistance profile in Eye Infections: A Global Concern with a retrospective Focus on a Large Hospital in north-Italy” explores the antibiotic resistance profile in eye infections in large hospitals in north Italy.

Minor Comments:

Introduction:

-              The more concise and targeted references to country of interest must be cited.

-              The article lacks a clean transition to the specific focus of the study for readers.

-              The complicated approach needed to cope with antibiotic resistance should be explained based on specific techniques or interventions connecting to the next sections.

Material and Method:

-              The section can be benefitted by providing additional information at the standards for sample inclusion and exclusion to make certain transparency and reproducibility.

-              The methodology phase must mention about moral issues or institutional review board approval, that's important for research concerning human topics or patient records.

Results:

-              The quick precis or interpretation of the key findings could assist readers understand the significance of the work.

-              The results section could gain from a discussion of significant findings, along with developments in resistance patterns or variations as compared to preceding research, to offer context and insights into the consequences of the results.

Discussion:

-              The dialogue effectively contextualizes the take a look at findings within the broader literature on antibiotic resistance in ocular infections. However, it could be bolstered by explicitly linking the observe findings to the prevailing understanding hole or studies query addressed by the study.

-              Explain obstacles on the translation of the outcomes and the generalizability of the findings.

Comments on the Quality of English Language

The English language is fine

Author Response

(The authors gave the same response as above.)

Reviewer 3 Report

Comments and Suggestions for Authors

This article collects eye samples from patients in a large hospital in Milan, Italy, analyzes antibiotic resistance in the samples, and understands the local use of antibiotics. It proposes that hospitals can develop effective antimicrobial drug management plans to reduce antibiotic resistance rates and improve patient prognosis. The article discusses how countries or regions can effectively manage infectious diseases and combat antibiotic resistance. However, this article still has some shortcomings:

1. The abstract does not provide a good summary of the entire text, and the research methods of this article can be added;

2. The abstract states that this article aims to study the current status of antibiotic resistance in global eye infections, and whether it is accurate to use a hospital's research situation as an example;

3. The word Isolationi in "2" is incorrect;

4. Whether the collection targets of swabs include patients in neonatology and ophthalmology, as well as outpatient patients, are they consistent with the title "Overview of antibiotic resistance in ophthalmic infections";

5. From Table 2, it can be seen that no broad-spectrum bacteria were found in Gram negative bacteria β- The report of positive lactamases (ESBLs) indicates that the incidence of methicillin resistance in Staphylococcus aureus is relatively low (4.5%), which should be seen from Table 1, and the incidence of methicillin resistance in Staphylococcus aureus (4.5%) is not listed in the table;

6. A "Conclusion" section can be added;

7. There are few references and inconsistent formats.

Author Response

(The authors gave the same response as above.)

Round 2

Reviewer 1 Report

Comments and Suggestions for Authors

Dear authors.

I can see that the suggestions I made in my previous review have been incorporated into the text. Now, I believe that the article better defines the theoretical framework, the discussion has improved substantially as well as the conclusions, which now reflect the results.

Reviewer 3 Report

Comments and Suggestions for Authors

The author answered my question, and I agree with it. The revised manuscript has been revised based on my suggestions. The quality of the revised manuscript has been improved. I am satisfied with this.